# Applications of the Microscale Thermophoresis Binding Assay in COVID-19 Research

**DOI:** 10.3390/v15071432

**Published:** 2023-06-25

**Authors:** Damian T. Nydegger, Jonai Pujol-Giménez, Palanivel Kandasamy, Bruno Vogt, Matthias A. Hediger

**Affiliations:** 1Department of Nephrology and Hypertension, Inselspital, University of Bern, Kinderklinik, Freiburgstrasse 15, 3010 Bern, Switzerland; damian.nydegger@unibe.ch (D.T.N.);; 2Department of Biomedical Research, Inselspital, University of Bern, Kinderklinik, Freiburgstrasse 15, 3010 Bern, Switzerland

**Keywords:** COVID-19, microscale thermophoresis, SLC6A20 amino acid transporter, antiviral agents

## Abstract

As the COVID-19 pandemic progresses, new variants of SARS-CoV-2 continue to emerge. This underscores the need to develop optimized tools to study such variants, along with new coronaviruses that may arise in the future. Such tools will also be instrumental in the development of new antiviral drugs. Here, we introduce microscale thermophoresis (MST) as a reliable and versatile tool for coronavirus research, which we demonstrate through three different applications described in this report: (1) binding of the SARS-CoV-2 spike receptor binding domain (RBD) to peptides as a strategy to prevent virus entry, (2) binding of the RBD to the viral receptor ACE2, and (3) binding of the RBD to ACE2 in complex with the amino acid transporter SLC6A20/SIT1 or its allelic variant rs61731475 (p.Ile529Val). Our results demonstrate that MST is a highly precise approach to studying protein–protein and/or protein–ligand interactions in coronavirus research, making it an ideal tool for studying viral variants and developing antiviral agents. Moreover, as shown in our results, a unique advantage of the MST assay over other available binding assays is the ability to measure interactions with membrane proteins in their near-native plasma membrane environment.

## 1. Introduction

Coronavirus disease 2019 (COVID-19) is caused by the severe acute respiratory syndrome coronavirus 2 (SARS-CoV-2). This enveloped positive-strand RNA virus, belonging to the group of β-coronaviruses that infect mammals, is characterized by club-like spike glycoproteins projecting from its surface [1,2]. The disease emerged in December 2019, with the first cases observed in Wuhan, China, followed by global spread, resulting in the largest pandemic so far this century [3,4,5]. Upon entry into the respiratory tract, one of the major sites of infection, SARS-CoV2 uses its spike protein receptor binding domain (RBD) to bind to the angiotensin-converting enzyme-2 (ACE2) receptor, which is located on the surface of lung epithelial cells. Binding is followed by priming of the spike protein through the peptidase TMPRSS2, which in turn triggers endocytosis and stimulates virus production and viral spread. The most common symptoms of COVID-19 are fever, dry cough, dyspnea, myalgia, and fatigue. Although SARS-CoV-2 infection can be asymptomatic, the disease is fatal in others, particularly the elderly or people who are immunocompromised [4,6]. As of April 2023, there have been more than 760 million confirmed cases and nearly 7 million deaths, according to the World Health Organization [7].

SARS-CoV-2 was the third outbreak of coronaviruses this century. Previously, severe acute respiratory syndrome coronavirus (SARS-CoV) emerged in 2002 and Middle East respiratory syndrome (MERS) in 2012. These repeated outbreaks highlight the importance of having reliable tools at hand to quickly investigate emerging viruses.

Here, we present and discuss the various applications of a binding assay known as microscale thermophoresis (MST) and review its suitability for coronavirus research. Briefly, MST depends on the movement of molecules in a temperature gradient (Figure 1). The rate of this movement is sensitive to changes in size, charge, and hydration shell. As a result of the binding of two molecules, the movement rate changes, which can be detected by MST. To measure changes in movement, one of the binders must be fluorescently labeled. To determine the binding affinity, a serial dilution of the ligand is generated and assessed in the capillaries where the binding experiments take place. For each capillary, the initial fluorescence is measured, then the infrared laser is activated to generate a local temperature gradient within the capillary (Figure 1a). This causes the labeled binder (e.g., labeled RBD) to move out of the focal area of the fluorescence detector, resulting in a decrease in fluorescence (Figure 1b). After switching off the IR laser, the labeled binder moves back into the focal area and the fluorescence returns to its initial state. A protein (e.g., RBD) that is bound to a ligand (e.g., a peptide) will exhibit a different rate of motion in response to laser exposure than an unbound protein because its size, charge, and/or hydration shell will change after binding. For each dilution, fluorescence is measured at the same time point, resulting in an affinity curve which is then used to determine the dissociation constant (K_D_) (Figure 1c) [8,9,10,11,12,13]. 

There are a variety of alternative biochemical assays that can be used to determine the dissociation constant (K_D_) of protein–ligand interactions such as thermal shift assay [14], surface plasmon resonance (SPR) [15], circular dichroism [14], isothermal titration calorimetry [16], small-angle X-ray scattering [17], and nuclear magnetic resonance spectrometry [18], among others. Given the wide variety of approaches that exist for affinity measurements, our goal was to select the method that offered the greatest advantages for our applications.

Importantly, the techniques mentioned above usually require purified proteins in relatively large quantities. But it is often costly to purchase these purified proteins, and the alternative of performing protein purification in-house can be challenging. In addition, membrane proteins often behave differently when extracted from their natural lipid environment. In this regard, a major advantage of the MST assay is that it allows measurements with crude cell lysates [19,20], thus avoiding harsh protein purification procedures and also greatly simplifying sample preparation. In addition, MST allows the retention of the natural lipid environment and potential protein binding partners. Thus, MST offers the possibility of performing affinity measurements in a native lipid bilayer environment, which distinguishes this approach from the other binding assays mentioned above.

In our study, three different MST approaches were evaluated as resources for coronavirus disease investigation. One approach was to use purified RBD protein and to determine the affinity between RBD and potential peptide ligands as a strategy to prevent viral RBD binding to ACE2. Another approach was to use cell lysates overexpressing the membrane protein receptor ACE2, which allows physiologically relevant measurements of the RBD-ACE2 interaction. In the third approach, we examined the effect of ACE2 interacting partners on RBD-ACE2 binding. For this purpose, we investigated the co-expression of ACE2 with amino acid transporter SLC6A20 or its allelic variant rs61731475 (p.Ile529Val).

For further information on this versatile MST binding assay, please refer to previous publications showing a variety of additional applications of this cutting-edge technique [9,10,11,12,21].

## 2. Materials and Methods

### 2.1. Labeling of Purified Proteins

RBD (recombinant SARS-CoV-2, S1 subunit protein) was obtained from RayBiotech, Inc., Peachtree Corners, GA (Lucerna-Chem #230-30162, Luzern, Switzerland) and labeled according to the protein labeling protocol of NanoTemper Technologies GmbH (Munich, Germany). The labeling protocol features the N-hydroxy succinimide (NHS) coupling of the fluorescent dye NT647 (RED-NHS 2nd Generation, NanoTemper # MO-L011). The fluorescence dye of the kit carries a reactive NHS-ester group that reacts with the primary amines (i.e., lysine residues) to form a covalent bond. The labeling and the subsequent MST experiments were performed in PBS-TR (PBS + 0.05% Triton X100).

### 2.2. Cell Culture and Lysate Preparation

A HEK293 cell line stably expressing pcDNA3-ACE2 (WT)-8his (Addgene # 149268; Addgene Europe, Teddington, UK) was generated using geneticin (g418) selection. Prior to MST measurements, cells were washed with PBS, removed with a cell scraper and collected by centrifugation. The resulting pellets were suspended in PBS-T (PBS + 0.05% Tween-20) and snap-frozen in liquid nitrogen for 1 min. This was followed by homogenization in a Teflon-glass homogenizer. Subsequently, the samples were centrifuged at 1500× *g* for 5 min and the upper milky fractions, which contained membranes and vesicles enriched in the overexpressed protein, were collected and used for His-tag labeling.

### 2.3. His-Tag Labeling of Cell Lysate

Labeling of cell lysates was performed according to the protocol of the His-tag labeling kit RED-tris-NTA 2nd Generation (NanoTemper # MO-L018). The optimal protein concentration for labeling the cell lysate was determined by titrating the cell lysate against a constant dye concentration (25 nM) on MST. The optimal protein concentration used corresponded to the concentration at which the curve began to saturate and was 0.56 mg/mL. As recommended in the protocol of NanoTemper, PBST-buffer (PBS + 0.05% Tween-20) was used for the labeling and measurements.

### 2.4. MST Measurements

The measurements were performed using the RED channel Monolith NT.115 MST device (NanoTemper Technologies GmbH) and the corresponding Monolith capillaries (NanoTemper; MO-K022). The data were collected with MO. Control v2 and evaluated using the MO. Affinity v2.3 Software (NanoTemper).

### 2.5. Generation of the SLC6A20 Variant I529V (Ile529Val; rs61731475)

For the co-expression experiments with the SLC6A20 amino acid transporter, we purchased the NM_020208 Myc-DDK-tagged-human ORF clone #RC215764 (OriGene; OriGene EU, Herford, Germany). The I529V variant was generated by site-directed mutagenesis using the standard polymerase chain reaction (PCR)-based approach, as previously described [22]. The primer used to generate the SLC6A20_I529V variant had the sequence 5′ CCTGAGCGACTACGTCCTCACGGGGACCC 3′.

### 2.6. Amino Acid Sequences of the RBD-Binding Peptides That Were Tested with MST

NB001R: RRRRRRFFERHHMVGSCMRAFHQL (24 Residues)

NB001: FFERHHMVGSCMRAFHQL (18 Residues)

NB002: FAHMNWKMQWLQKWQQGK (18 Residues)

## 3. Results

SARS-CoV-2 relies on the receptor ACE2 to enter cells of the human body such as the epithelial cells of the lung. The first contact between the virus and the target cell is the binding of the RBD of the viral spike protein to the extracellular catalytic domain of ACE2. This entry pathway is an attractive target for the development of antiviral agents that could bind either the RBD of the viral spike protein or its receptor ACE2 to prevent viral infection. Given that cell membranes are complex structures, the binding between RBD and ACE2 is expected to be influenced by a number of factors. These include interacting proteins, including the amino acid transporter SLC6A20, which is being investigated in our laboratory.

In the present study, we present three examples to illustrate how MST can be employed to measure protein–ligand interactions that are relevant in COVID-19 research (Figure 2).

### 3.1. Determination of the Binding of SARS-CoV-2-S1-RBD to Antiviral Agents

Since the crystal structure of SARS-CoV-2 RBD bound to ACE2 is known [23,24], it is now possible to design peptides that could prevent the binding of the virus to ACE2. One strategy is to engineer peptides that can bind to the receptor binding domain of the viral spike protein in such a way that the ACE2 binding site is occupied and the virus cannot enter the cell. In silico peptide engineering can provide putative RBD-binding peptides, but binding needs to be validated experimentally, for which we used MST as a suitable assay for peptide screening. To confirm the applicability of MST for this purpose, a series of putative RBD-binding peptides were tested with MST. The peptides were kindly provided by Reymond and Gunasekera (see Acknowledgments), who had developed them using in silico methods. With these experiments, we aimed to validate the binding efficacy of the designed peptides to RBD, which is a first step towards assessing whether these peptides could prevent SARS-CoV-2 infection.

When working with nonfluorescent purified proteins (e.g., the spike protein), there are the following two labeling choices: (1) the unspecific labeling procedure, whereby the RED-NHS 2nd generation dye binds randomly to primary amides (i.e., lysine residues), and (2) the specific His-tag labeling procedure which involves labeling to the specific His-tag site of the expressed protein of interest. Both labeling methods are well established.

Nonspecific labeling of RBD was used in order to test the binding of peptides to RBD (Figure 2a). As expected from the in silico predictions, our MST results (Figure 3) confirm the binding of peptide NB001R to the spike protein, with a K_D_ of 2.08 μM (±0.43 μM) (Figure 3a). The measurements for the different dilutions tested are shown in Figure 3b. They are based on the MST traces (i.e., the IR-triggered fluorescence changes) shown in Figure 3c. Similarly, the binding of two other peptides, NB001 and NB002, predicted to bind RBD, were tested using MST and binding was confirmed (NB001, K_D_ = 1.08 μM (±0.51 μM) and NB002, K_D_ = 0.94 μM (±0.54 μM) (Appendix A)). This demonstrates that MST is a suitable tool for screening the binding ability of peptides to RBD.

### 3.2. Determination of the Binding of SARS-CoV-2-S1-RBD to His-Tag-Labeled ACE2

While binding studies of purified proteins are an appropriate strategy to assess the interaction of viral spike proteins with potential ligands, a more sophisticated strategy is required when the ligand of interest is a transmembrane protein such as ACE2. Remarkably, in our studies, MST allowed us to assess the binding of transmembrane proteins to ligands using crude cell lysates. This success is likely due to the fact that the natural lipid environment of the proteins under study is largely preserved in our crude cell lysate preparations. For this type of experiment, nonspecific labeling of the protein of interest would not be a reliable option because it would label any lysine-containing proteins present in the cell lysate. An alternative approach would be to generate a cell line that overexpresses a recombinant GFP or RFP version of the protein of interest. However, the insertion of a bulky fluorescent tag could interfere with the expression or function of the target protein. For that reason, we preferred the His-tag labeling approach. For this purpose, a stable HEK293 cell line overexpressing ACE2 containing a His-tag attached to the Cterminus was generated, and the His-tag dye was mixed directly with the cell lysates for labeling. A serial dilution of RBD was added to the different capillaries prior to the MST measurements. As shown in Figure 4a, the K_D_ of membrane-embedded ACE2 to RBD was 37.2 nM (±10.7 nM). A representative dose–response curve is shown in Figure 4b and MST traces are shown in Figure 4c. Our results demonstrate that this approach allows for accurate and reliable binding measurements in the natural lipid environment without the need to prepare purified protein.

### 3.3. Determination of the Binding of SARS-CoV-2-S1-RBD to the ACE2-SLC6A20 Heterodimeric Complex in the Native Lipid Bilayer Environment

Often, proteins in the cell membrane are not expressed as monomers but form homomultimers or heteromultimers with other proteins. It is well-known that complexation of proteins can alter the ability of protein monomers to interact with their environment [25,26,27]. Since proteins are usually purified as monomers, this aspect cannot be investigated using a conventional binding procedure, as ACE2 forms a complex with SLC6A20 (Figure 5c). However, MST allows working with crude membrane extracts in which a native lipid environment is present and protein complexes are still intact, making it an ideal tool to overcome this limitation. This protein complex is of particular interest because SLC6A20 has been shown to have an impact on the outcome of SARS-CoV2 infection [28,29]. In addition, variants of SLC6A20 are associated with diabetes mellitus [30], which we believe may also impact the infection efficiency [31]. And recently, the SLC6A20 variant I529V (Ile529Val; rs61731475) was indeed reported to affect the SARS-CoV-2 clinical outcome in Italian families [32].

To assess whether the co-expression of SLC6A20 and/or variant I529V together with ACE2 affects RBD binding, we wanted to investigate whether our MST method could answer this question. For this experiment, the stable cell line expressing ACE2 was transfected with either an empty vector (control), SLC6A20_WT, or SLC6A20_I529V. As described previously, measurements were performed with cell lysates after His-tag labeling. The results were normalized to the control condition (empty vector) to determine the fold change of K_D_ for complexes of ACE2 with SLC6A20_WT and SLC6A20_I529V. Interestingly, as shown in Figure 5a, the ACE2-SLC6A20_WT complex exhibited slightly stronger binding to RBD compared to the control, with a fold change of 0.67 (±0.12), whereas for the ACE2-SLC6A20_I529V complex there was somewhat weaker binding, with a fold change of 2.03 (±0.33). The representative dose–response curves and MST traces are shown in Figure 5b.

These experiments demonstrate that MST enables the study of protein–ligand interactions in the native lipid bilayer environment, including intact protein complexes, and thus provides in-depth insight into the binding behavior. Of course, the binding of ACE2 to RBD, as shown in Figure 5c, is only the first step of a complex infection process. Therefore, further studies are still required using different approaches, such as experiments with pseudoviruses or the original Wuhan SARS-CoV-2 virus and its variants, to assess whether SLC6A20 or its genetic variants affect the outcome of SARS-CoV-2 infection.

## 4. Discussion

In our study, we present several applications of MST that illustrate the versatility of this approach in coronavirus research and subsequent drug discovery: Binding of peptides to RBD that may ultimately prevent virus entry, and binding of RBD to ACE2, either alone or in complex with the amino acid transporter SLC6A20 or its allelic variant SLC6A20_I529V.

The K_D_ of membrane-embedded ACE2 for RBD was measured to be 37.2 nM (±10.7 nM). This value falls within a comparable range to the reported value (44.2 nM) based on measurements using the SPR assay [24]. Although similar results were obtained with both methods, we believe that MST has distinct advantages: While in SPR, purified RBD is immobilized on a sensor chip and the ligand, which is purified ACE2, is added at various concentrations to perform the measurements [24]; in MST, purified RBD is used for the dilutions and there is no need for purified ACE2, nor is the step of immobilizing RBD required. Thus, crude membrane extracts of a cell line transiently or stably expressing ACE2 can be used for the MST experiments without the need for further purification. His-tag labeling is performed within 30 min, and if ACE2-GFP was used as an alternative, no labeling was needed at all. Furthermore, working with crude cell lysates is more cost and time effective. In addition, as mentioned earlier, MST measurements were performed in the native environment of crude membrane extracts, where the associated lipids and proteins are likely still embedded in the membrane after cell rupture, allowing binding affinity to be measured at conditions closer to in vivo, which is in sharp contrast to the SPR method. In this context, our co-expression studies have shown that the presence of protein partners such as SLC6A20 can affect the binding affinity of ACE2 for RBD. Therefore, the absence of the additional partners present in the natural environment of ACE2 could distort the interpretation of the physiological significance of measurements using only purified proteins, which further highlights the advantage of the MST approach over other binding assays.

Because MST allows for the reliable examination of ligand interactions with membrane proteins in complex with other proteins, we were able to test whether the ACE2-SLC6A20-WT complex alters the binding affinity of RBD to ACE2 compared with ACE2 alone. Indeed, our experiments demonstrate that MST is capable of distinguishing subtle changes in binding affinity, which could be key to interpreting changes in the infectivity rate due to alterations in the ACE2-SLC6A20 complex. Since our experiments in fact show different K_D_ values for WT and I529V, this supports our hypothesis of a possible effect of this mutation on the infectivity of the SARS-CoV2 capacity, as previously hypothesized [28,29].

Interestingly, in contrast to our results, a recent study showed lower RBD binding for the ACE2-SLC6A20_WT complex with a higher K_D_ value of 63.23 nM [33], whereas the K_D_ value for ACE2 alone (43.64 nM) was fairly consistent with the K_D_ value obtained in our study by MST under the same conditions (37 nM, see Figure 4). The difference in ACE2-SLC6A20_WT binding may be attributed to the different binding method used in the latter study, namely flow cytometry. During measurement and incubation with RBD, the cells are still intact in flow cytometry, which means that only ACE2 expressed at the membrane can bind RBD. Interestingly, the co-expression of ACE2 and SLC6A20_WT was shown to result in a significant decrease in surface expression of the ACE2-SLC6A20-WT complex compared with ACE2 alone in the latter study. Consequently, a dramatic 2.1-fold decrease in maximal binding strength was observed for the ACE2-SLC6A20-WT complex. While K_D_ values should be independent of maximal binding levels, a lower signal-to-noise ratio could hinder proper curve fitting and K_D_ determination, which in this case could potentially explain the differences in K_D_ between MST and flow cytometry methods. In contrast to flow cytometry, our MST measurements showed similar maximal binding levels for all overexpressed constructs (Figure 5). This is likely a consequence of using cell lysates for MST that contain a mixture of surface membrane and cytosolic components, whereas flow cytometry relies only on proteins expressed on the surface membrane. Finally, it is worth mentioning that, in the same study, the K_D_ for the binding of the ACE2 N-terminal peptidase domain to RBD was also measured via biolayer interferometry, yielding a K_D_ value of 18.4 ± 0.03 nM, which is lower than that determined by MST or flow cytometry. This result once again highlights the differences that might arise as a consequence of using purified proteins instead of proteins in their native cellular environment in the binding study.

It is also worth noting that another amino acid transporter member of the SLC6 family, SLC6A19, forms a complex with ACE2 in enterocytes of the small intestine. There, SLC6A19 and its genetic variants could influence SARS-CoV-2 infectivity across the intestinal barrier [23,34,35]. Further experiments are needed to clarify whether and how SLC6 amino acid transporters and their allelic variants affect COVID-19 infectivity in different epithelial tissues.

The strength of MST in virus research is further evidenced by several recent studies. In one of them, the nonstructural protein 9, an RNA-binding protein essential for viral replication of SARS-CoV, was the subject [36]. While we measured protein–protein and peptide–protein interactions, the affinity of a protein–ssDNA interaction was determined in this study. This further highlights the broad applicability of MST. Another report shows the protein–protein interaction of nonstructural protein 15, expressed in MERS, with other nonstructural proteins [37]. Recently, a paper was published in which MST was used to screen pan-coronaviral major protease inhibitors. Similar to our initial experiments, but on a much larger scale, in silico experiments revealed promising inhibitors and confirmed binding affinity with MST [38].

In summary, it can be concluded that, based on our findings on membrane proteins, MST is a straightforward method to detect protein–protein and protein–peptide interactions based on changes in molecular weight and hydration shell. While the MST method is known to be well-suited for measuring the interactions of purified soluble proteins, for membrane proteins with alternative hydrophobic and hydrophilic domains, purification of the required amounts of high-quality concentrated material is challenging, and the resulting protein–detergent complexes compromise the binding events. Therefore, the strength of our MST approach lies in its ability to perform measurements in crude cell lysates where membrane proteins can retain their near-natural environment. Moreover, the influence of co-expressed membrane proteins that form complexes with ACE2 such as SLC6A20 can be readily studied using our MST approach. The successful use of MST with crude membrane extracts has also been demonstrated for other SLC solute carriers such as the H^+^-coupled oligopeptide transporter PepT1/SLC15A1 [19] and the lysosomal SLC15A4 peptide/histidine transporter (Hediger et al., unpublished data). This highlights the versatility of the MST approach for studying membrane proteins, even when expressed in intracellular membranes. Moreover, the herein-presented approach is likely applicable to other coronaviruses, including emerging virus variants.

## Figures and Tables

**Figure 1 viruses-15-01432-f001:**
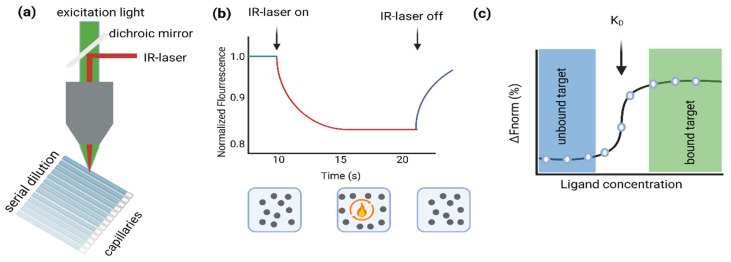
(**a**) Schematic representation of the MST assay. (**b**) (Upper panel.) Representative MST trace showing fluorescence measurement upon IR stimulation (On/Off) over the indicated time period. (Lower panel.) Schematic representation of the response of the proteins in solution upon IR stimulation. (**c**) Representative fitting of the fluorescent values obtained for the different ligand concentrations at a given time point of the MST traces. The figure was created using the BioRender software (https://www.biorender.com).

**Figure 2 viruses-15-01432-f002:**
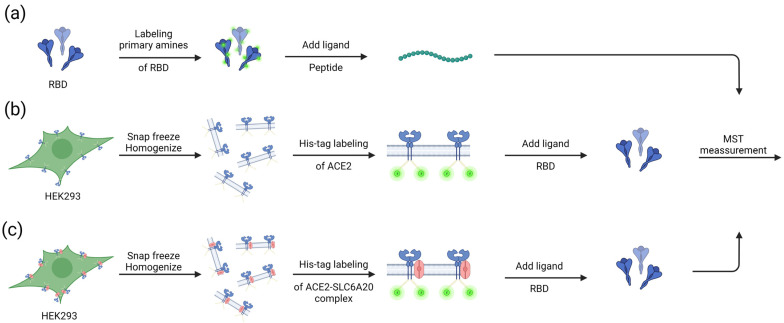
Applications of MST in coronavirus research. (**a**) The primary amines of purified RBD were labeled and a serial dilution of peptides that were predicted to bind RBD were added to perform MST. (**b**) C-terminal His-tag ACE2 is overexpressed in HEK293 cells, the ACE2 in the cell lysate was labeled with His-tag dye, and a serial dilution of RBD was added to perform MST. (**c**) His-tagged ACE2 and SLC6A20 were overexpressed in HEK293 cells, the ACE2 in the cell lysate was tagged with His-tag dye, and a serial dilution of RBD was added to perform MST. The figure was created using the BioRender software (https://www.biorender.com).

**Figure 3 viruses-15-01432-f003:**
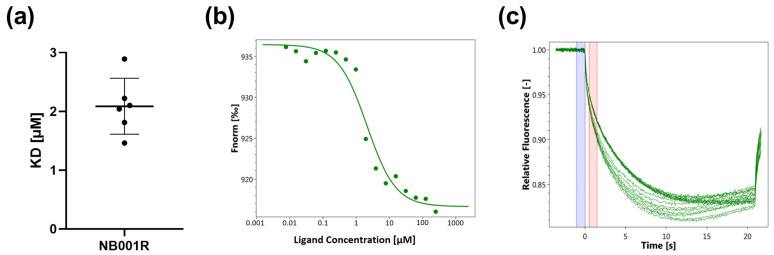
Binding of SARS-CoV-2-S1-RBD to peptide NB001R. (**a**) The K_D_ of NB001R and RBD is 2.08 μM. A serial dilution of the peptide from 1 mM–30 nM was used. (**b**) Representative dose–response curve. The red bar indicates the selected time point for the dose response used to determine the K_D_ (**c**) MST traces.

**Figure 4 viruses-15-01432-f004:**
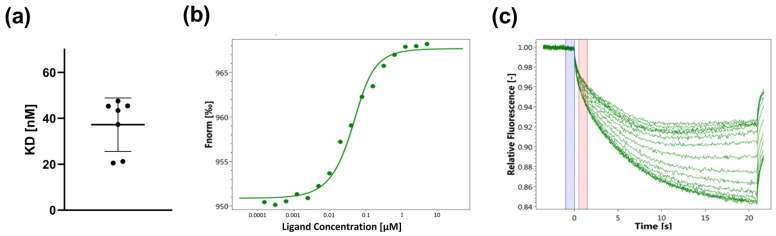
Binding of SARS-CoV-2-S1-RBD to His-tag-labeled ACE2. (**a**) The K_D_ of RBD and ACE2 in its natural lipid environment is 37 nM. A serial dilution of RBD from 5 μM–0.15 nM was used. Each K_D_ value was calculated from dose–response curves obtained in independent experiments using the stably transfected HEK293 ACE2 His-tag cell line. (**b**) Representative dose–response curve. (**c**) MST traces.

**Figure 5 viruses-15-01432-f005:**
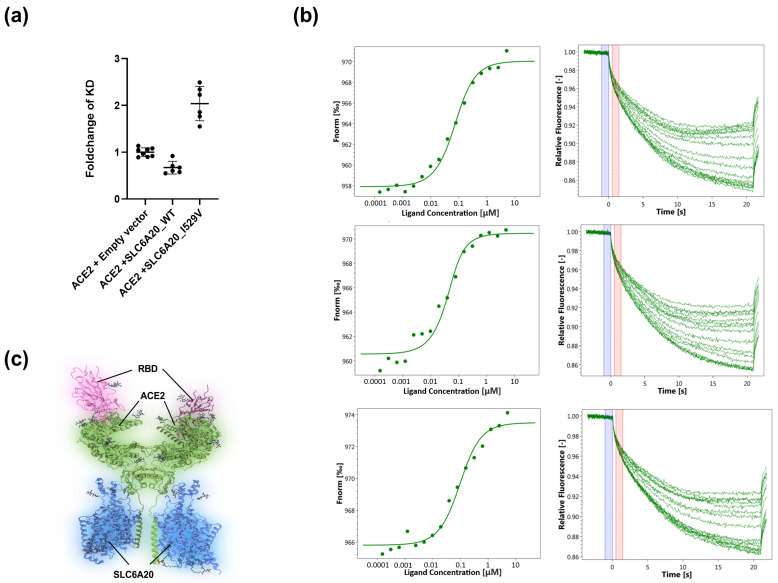
Binding of SARS-CoV-2-S1-RBD to the ACE2-SLC6A20 heterodimeric complex. (**a**) Impact of SLC6A20_WT and SLC6A20_I529V on the binding of ACE2 to RBD. The data shows a stronger binding of RBD to ACE2-SLC6A20 WT. The SLC20_I529V variant showed a reduction in binding affinity. A serial dilution of RBD from 5 μM to 0.15 nM was used. Three biological replicates with at least two technical replicates each were performed. (**b**) Dose response and MST traces. Top: ACE2 + empty vector; middle: ACE2 + SLC6A20_WT; bottom: ACE2 + SLC6A20_I529V. (**c**) Structure of ACE2-SLC6A20 complex binding to RBD based on PDB ID: 7Y75 (https://www.rcsb.org/structure/7Y75) (Shen, Y.P., Li, Y.N., Zhang, Y.Y., Yan, R.H., [33]). The figure was created using the BioRender software (https://www.biorender.com).

## Data Availability

All available data are included in the Section 3 and Appendix A.

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
