# Peer review of "Applications of the Microscale Thermophoresis Binding Assay in COVID-19 Research"

_viruses, 2023, doi:10.3390/v15071432_

Round 1

Reviewer 1 Report

In this communication, the microscale thermophoresis, a relatively new approach, was used to measure binding affinity of SARS-CoV-2 RBD with 1) peptides, 2) ACE2 and 3) ACE2 in complex with SLC6A20/SIT1 or its variant. The results showed that MST is a reliable and versatile tool for protein-protein and/or protein-ligand interactions analysis. More importantly, the unique advantage of the MST assay over other assays is that the binding affinity can be measured in crude cell lysates where the membrane protein or protein complex likely retains its natural structure in near-native membrane environment, which will contribute to the study of SARS-CoV-2 entry and development of entry inhibitors. Below are some concerns and suggestions.

1) Line 125, 0.56 mg/μl is a very high concentration. Please double check.

2) Is there any article about the peptides (NB001R, NB001 and NB002) published in journals for citation?

3) Occasionally, peptide binds to protein nonspecifically. Do the authors have enough funding to purchase NTD of Spike protein to see whether these three peptides bind NTD or not.

4) Dose each data point in Fig 4a and 5a represent result from independent transfection or from technical replication of one transfection?

Reviewer 2 Report

The work by Nydegger et al., describing the Applications of the microscale thermophoresis binding assay 2 in COVID-19 research is overall an interesting study. While the authors do point out several advantages of using MST over classical techniques like SPR in COVID-19 research, papers have already been published in this aspect: (PMID: 35337106), (PMID: 30135128), (PMID: 29925659); the authors should also cite these papers. 
1. The authors focus on ACE 2 which has not been done earlier but is the method sensitive enough to detect minor differences in binding affinity between ACE proteins of different variants.

2. Also , I believe the authors want to highlight the use of MST with proteins being still in natural environment, to increase the impact further authors should show a couple of other example of membrane protein to be able to generalize it.

Minor language corrections are needed.

Reviewer 3 Report

This manuscript by Damian Nydegger and co-workers illustrates the evaluation of three different microscale thermophoresis (MST) approaches as means for coronavirus disease study. MST is a powerful new method for the quantitative analysis of protein-protein interactions with low sample consumption. The first approach utilizes purified RBD protein and determines its affinity to potential peptide ligands as a strategy to prevent viral binding of RBD to ACE2. Another approach uses overexpressing ACE2 HEK293 cell lysates to measure the interaction between RBD and ACE2. In the third approach ACE2 is co-expressed with SLC6A20, an amino acid transport protein able to form a protein complex with ACE2 on the plasma membrane of lung epithelial cells, to examine the effect of protein expression partners on RBD-ACE2 binding. 

I have the following suggestions and comments.

1. Lines 34-37:

Please substitute “This virus belongs to the group of β-coronaviruses that infect mammals, are enveloped by spike glycoproteins, and have a positive-sense single-stranded RNA genome”.  with: This enveloped positive-strand RNA virus, belonging to the group of β-coronaviruses that infect mammals, is characterize by club-like spike glycoproteins projecting from its surface.

2. Lines 41-43

Please substitute “Binding is followed by priming of the spike protein through the peptidase TMPRSS2, which in turn triggers endocytosis and viral production”. with:“Binding is followed by priming of the spike protein through the peptidase TMPRSS2, which in turn triggers endocytosis and stimulates viral spread”.

3. References

I would suggest adding other papers, such as those listed below, to make the Introduction even more complete.

Jerabek-Willemsen M, André T, Wanner R, Roth HM, Duhr S, Baaske P, Breitsprecher D. MicroScale Thermophoresis: Interaction analysis and beyond, Journal of Molecular Structure. 2014. 1077, 101-113, https://doi.org/10.1016/j.molstruc.2014.03.009.

Seidel SA, Dijkman PM, Lea WA, van den Bogaart G, Jerabek-Willemsen M, Lazic A, Joseph JS, Srinivasan P, Baaske P, Simeonov A, Katritch I, Melo FA, Ladbury JE, Schreiber G, Watts A, Braun D, Duhr S. Microscale thermophoresis quantifies biomolecular interactions under previously challenging conditions. Methods. 2013 Mar;59(3):301-15. doi: 10.1016/j.ymeth.2012.12.005. 

Romain M, Thiroux B, Tardy M, Quesnel B, Thuru X. Measurement of Protein-Protein Interactions through Microscale Thermophoresis (MST). Bio Protoc. 2020 Apr 5;10(7):e3574. doi: 10.21769/BioProtoc.3574. 

4. The language of the manuscript must be improved. 

The language needs improvement. 

Round 2

Reviewer 1 Report

The manuscript improved after revision.

Reviewer 2 Report

The authors have addressed all my concerns

It is acceptable